# Vitisin A, a Resveratrol Tetramer, Improves Scopolamine-Induced Impaired Learning and Memory Functions in Amnesiac ICR Mice

**DOI:** 10.3390/biomedicines10020273

**Published:** 2022-01-26

**Authors:** Lih-Geeng Chen, Ching-Chiung Wang, Yi-Shan Lee, Yi-Yan Sie, Chi-I Chang, Wen-Chi Hou

**Affiliations:** 1Department of Microbiology, Immunology and Biopharmaceuticals, College of Life Sciences, National Chiayi University, Chiayi 600, Taiwan; lgchen@mail.ncyu.edu.tw; 2Traditional Herbal Medicine Research Center, Taipei Medical University Hospital, Taipei 110, Taiwan; crystal@tmu.edu.tw; 3Graduate Institute of Pharmacognosy, Taipei Medical University, Taipei 110, Taiwan; m303103009@tmu.edu.tw; 4School of Pharmacy, College of Pharmacy, Taipei Medical University, Taipei 110, Taiwan; 5Ph.D. Program in Clinical Drug Development of Herbal Medicine, College of Pharmacy, Taipei Medical University, Taipei 110, Taiwan; d339110003@tmu.edu.tw; 6Department of Biological Science and Technology, National Pingtung University of Science and Technology, Pingtung 912, Taiwan

**Keywords:** acetylcholinesterase, brain-derived neurotrophic factor (BDNF), scopolamine, vitisin A

## Abstract

Resveratrol has been reported to exhibit neuroprotective activities in vitro and in vivo. However, little is known about resveratrol tetramers of hopeaphenol, vitisin A, and vitisin B with the same molecular mass in the improvement of degenerative disorders. In this study, two 95% ethanol extracts (95EE) from stem parts of *Vitis thunbergii* Sieb. & Zucc. (VT-95EE) and from the root (R) parts of *Vitis thunbergii* var. *taiwaniana* (VTT-R-95EE) showed comparable acetylcholinesterase (AChE) inhibitory activities. It was found that VT-95EE and VTT-R-95EE showed different distribution patterns of identified resveratrol and resveratrol tetramers of hopeaphenol, vitisin A, and vitisin B based on the analyses of HPLC chromatographic profiles. The hopeaphenol, vitisin A, and vitisin B, showed AChE and monoamine oxidase-B inhibitions in a dose-dependent manner, among which vitisin B and vitisin A exhibited much better activities than those of resveratrol, and had neuroprotective activities against methylglyoxal-induced SH-SY5Y cell deaths. The scopolamine-induced amnesiac ICR mice treated with VT-95EE and its ethyl acetate-partitioned fraction (VT-95EE-EA) at doses of 200 and 400 mg/kg, or vitisin A at a dose of 40 mg/kg, but not vitisin B (40 mg/kg), were shown significantly to improve the impaired learning behaviors by passive avoidance tests compared to those in the control without drug treatments (*p* < 0.05). Compared to mice in the control group, the brain extracts in the vitisin A-treated mice or donepezil-treated mice showed significant reductions in AChE activities and malondialdehyde levels (*p* < 0.05), and elevated the reduced protein expressions of brain-derived neurotrophic factor (BDNF) and BDNF receptor, tropomyosin receptor kinase B (TrkB). These results revealed that vitisin A was the active constituent in the VT-95EE and VTT-95EE, and the VT medicinal plant and that the endemic variety of VTT has potential in developing functional foods for an unmet medical need for neurodegenerative disorders.

## 1. Introduction

Dementia is a global pandemic and is ranked as one of top 10 causes of worldwide death in the recent ten years [1]. The Alzheimer’s disease (AD) accounted for on average 70% all dementia populations [2]: those who are characterized by memory lapses to short-term memory impairments, dysfunctions in expressing thoughts and performing routine tasks, and spatial awareness dysfunctions, and finally the loss of ability to participate the general activities [3]. The developments of acetylcholinesterase (AChE) inhibitors based on cholinergic hypothesis, such as rivastigmine, donepezil, tacrine, and galantamine, are only approved as therapeutic drugs for AD medical management, and reports revealed that AChE inhibitors exhibited short-term improvement in cognition dysfunctions [4,5,6,7]. The cholinergic enzyme of AChE is responsible for acetylcholine hydrolysis to terminate neuronal transmission. The acetylcholine levels were associated with learning and memory in the hippocampus, and the neurodegenerative disorders in general showed the failure of cholinergic circuitry in the basal forebrain, which were responsible for the cognition dysfunctions [8,9]. The deficits in the cholinergic neuron were generally found in the normal aging and in AD patients; in those who showed the classic symptoms of cognitive declines [10,11], therefore, acetylcholine was recognized as the decider between encoding mode and retrieval mode in memory processing. Scopolamine, also called hyoscine, is an anticholinergic drug that is used to treat motion sickness [12]. Scopolamine is a competitive muscarinic receptor antagonist, which is used to induce a reproducible, transient impairment in healthy animals and non-diseased humans [13], and which blocks acetylcholine effects within the nervous system to induce amnesia (or impaired memories), and this short-term effect could be restored by treating AChE inhibitors. Therefore, the scopolamine induction in vivo was generally used to investigate effects of AChE inhibitors on AD managements [14,15]. It has been reported that alkaloids were intensively investigated for their AChE inhibitory activities [16]. The FDA approved AChE inhibitor of galantamine, an alkaloid isolated from the bulbs and flowers of *Galanthus* spp of Amaryllidaceae, for the treatment of mild to moderate AD in 2001. The sanguinine, isolated from *Eucharis grandiflora* with a galanthamine skeleton, showed more potent AChE inhibition (IC_50_ of 0.1 μM) compared to that of galanthamine (IC_50_ of 1.07 μM) [17,18]. The hydrolysable tannins, such as tellimagrandin I, tellimagrandin II, and 1,2,3,6-tetra-*O*-galloyl-β-D-glucopyranoside, showed AChE inhibitory activities with IC_50_ of 18.4, 18.6, and 84.2 μM, respectively [19]. The demethylcurcumin, a minor component of natural curcuminoids, showed better AChE inhibitory activity than that of curcumin with IC_50_ of 66.94 μM [20].

“The free radical theory of aging” describes the central roles of reactive oxygen species (ROS) in the aging process, and are also reported to participate aging, cardiovascular diseases, and neurodegenerative diseases [21,22]. In cells, the main radical or ROS generations are the electron transport chains in mitochondria for energy productions; enzymatic metabolic processes, such as dopamine or 6-hydroxydopamine catalyzed by monoamine oxidase (MAO)-B or hypoxanthine/xanthine metabolisms catalyzed by xanthine oxidase; and non-enzymatic glycation of proteins. It is found that the increased MAO-B activity was correlated with age and had a positive correlation between the progress of AD in the transgenic mice and the brain’s MAO-B activity [23]. Therefore, inhibition of MAO-B activity may be a strategy to delay neurodegenerative disorders and improve quality of life in the elderly. The non-enzymatic glycation includes the formation of protein adducts via the nucleophilic attacks of aldehyde group in open-chain reduced saccharides and the metabolized intermediates of glycolysis, such as methylglyoxal (MGO). Finally the advanced glycation endproducts (AGEs) are generated, which impair protein or enzyme functions and also couple with the receptors for AGEs (RAGE) to promote ROS productions [24]. MGO is a major cell-permeant precursor of AGEs, and is reported to link with diabetes, aging and neurodegenerative diseases [25]; the glycated proteins might accelerate amyloid β peptide accumulations and consequently increase oxidative stress in neuronal cells [26]. The 5β-carboxystrictosidine and chlorogenic acid isolated from *n*-butanol fractions of *Uncaria hirsute* extracts exhibited neuroprotective activities to elevate cell viabilities against 6-hydroxydopamine-induced cell apoptosis in differentiated PC12 cells via the reductions of intracellular ROS levels [27]; the demethylcurcumin treatments showed neuroprotective activities to increase cell viabilities in the 6-hydroxydopamine-induced SH-SY5Y neuronal cell death [20]; the thiol-contained antioxidant peptides derived from yam tuber storage protein showed protective activities in the human umbilical vein endothelial cells toward MGO-induced cell death [28].

The medicinal plant of *Vitis thunbergii* Sieb. & Zucc. (VT) has long been whole plant uses as folk herbs in Taiwan to treat arthritis, diarrhea, hepatitis, and jaundice, which the isolated resveratrol oligomers and derivatives from root ethanol extracts have been reported to exhibit radical scavenging activities and inhibitions against platelet aggregations, including ampelopsin C, miyabenol A, (+)-ε-viniferin, (−)-viniferal, vitisinol A–G, (+)-vitisin A and (+)-vitisin C [29]. It has been reported that the treatments of the enriched fraction of VT ethanol extracts, the identified major components included (+)-vitisin A and (−)-vitisin B (resveratrol tetramer), and ampelopsin C (resveratrol trimer), exhibited anti-bone loss in the ovariectomized mice, and the active components of ampelopsin C and (−)-vitisin-B showed to promote osteoblastogenesis in differentiated primary mouse bone marrow cells [30]. The endemic variety of VT in Taiwan of *Vitis thunbergii* var. *taiwaniana* (VTT) is the so-called “small-leaf grape” with smaller leaves and fruits in phenotypes compared to those of the edible grape (*Vitis vinifera*). The small fruit of VTT is not used for eating in general. However, the dried, non-fermented leaves of VTT are materials used as substituents for tea leaves in Taiwan. The VTT extracts and the isolated resveratrol oligomers have been reported to have improved functions against metabolic syndromes in vitro and in vivo. The VTT ethanol extracts from stem parts (S) and root parts (R), but not leaf parts(L), exhibited anti-angiotensin converting enzyme (ACE) activities, and the VTT-S oral administrations was shown to reduce systolic blood pressure in spontaneously hypertensive rats, in which the isolated three resveratrol oligomers of (+)-vitisin A, ampelopsin C, and (+)-ε-viniferin showed anti-ACE activities in a dose-dependent manner better than that of resveratrol [31]. The oral administration of VTT-R was shown to reduce the weight gains in one-month high-fat diet-induced obese mice under the same feed intakes. Isolated resveratrol tetramers of (+)-vitisin A might be the active components to reduce the weight gains in 36-day high-fat diet-induced obese mice [32]. The VTT-S treatments were shown to improve the impaired blood glucose in obese Wistar rats by pre-induction of high-fat diet, which might be via inhibitions against dipeptidyl peptidase IV and α-glucosidase activities [33]. In this study, the enriched resveratrol oligomers of extracts of VT and its endemic variety of VTT, and the active component of vitisin A were investigated under in vitro and in vivo animal experiments; their potential use may be developed as functional foods and/or lead compounds for delaying the onset of degenerative disorders.

## 2. Experimental Section

### 2.1. Chemical and Reagents

The acetylthiocholine iodide, anti-mouse IgG, anti-rabbit IgG, deprenyl, 5′,5′-dithiobis(2-nitrobenzoic acid) [DTNB], dimethylsulfoxide (DMSO), kynuramine dihydrobromide, MGO, phosphate-buffered saline (PBS), sulforhodamine B (SRB), recombinant human MAO-B, were from Sigma Chemical Co. (St. Louis, MO, USA). The 2-thiobarbituric acid (TBA) and electrophoretic reagents were purchased from E. Merck Inc. (Darmstadt, Germany). The donepezil hydrochloride was purchased from Tokyo Chemical Industry Co. (Tokyo, Japan). The bovine serum albumin (BSA) and BCA ^TM^ protein assay kit were purchased from Thermo Fisher Scientific Inc. (Rockford, IL, USA). The recombinant human AChE (P22303) was from R&D Systems Inc. (Minneapolis, MN). The anti-brain-derived neurotrophic factor (anti-BDNF) antibody (GTX30088) and anti-tropomyosin receptor kinase B (anti-TrkB) antibody (GTX54857) were from GeneTex Inc. (Irvine, CA, USA).

### 2.2. Preparations of Crude Extracts and Partitioned Fractions of VT and VTT

For crude extracts of VT herbal plants, the only stem part was used as follows. The stem parts of VT were dried in an oven at 50 °C for 3 days, powdered, and passed through a 10 mesh sieve (May Chun Co. Ltd., Taichung, Taiwan) for further use. The powdered VT samples (14 kg) were soaked in 95% ethanol (W/V = 1:5) for 7 days at room temperature, the residues were soaked again in the repeated procedure. The combined extracts were evaporated and concentrated under reduced pressure to yield 95% ethanol extract (VT-95EE, 412.3 g). For hot-water extraction, the powdered VT samples (5.5 kg) were extracted with 18-L boiling water for 3 h, and the extract was filtered and evaporated and concentrated under reduced pressure below 45 °C, which were then freeze-dried to get hot-water extract (VT-HWE, 356 g) and stored at −20 °C for further use. For the VT-95EE fraction preparations, the 412.3 g VT-95EE was partitioned successively with ethyl acetate (EA) and *n*-butanol (BuOH) to obtain VT-95EE-EA (83.5 g), VT-95EE-BuOH (117.7 g), and VT-95EE-H_2_O (195.2 g). For crude extracts of different parts of VTT herbal plants, the root (R), stem (S), and leaf (L), were prepared according the previous report [32]. In brief, the different parts of VTT were dried in an oven at 50 °C for 3 days, powdered, and passed through a 10 mesh sieve (May Chun Co. Ltd., Taichung, Taiwan). Each powdered sample of VTT-R, VTT-S, and VTT-L was soaked in 95% ethanol (W/V = 1:5) for 7 days at room temperature; the residues were soaked again in the same procedure. The combined extracts were evaporated and concentrated under reduced pressure to yield 95% ethanol extract (VTT-95EE). For hot-water extraction, each powdered sample of each plant part (100 g each) was individually extracted with 1-L boiling water for 3 h, and the extract each was filtered, evaporated and concentrated under reduced pressure below 45 °C; they were then freeze-dried to get hot-water extracts (VTT-R-HWE, VTT-S-HWE, and VTT-L-HWE) and stored at −20 °C for further use. For the VTT-R-95EE fraction preparations, the 665.3 g VTT-R-95EE was partitioned successively with ethyl acetate (EA) and *n*-butanol (BuOH) to obtain VTT-R-95EE-EA (507 g), VTT-R-95EE-BuOH (20.7 g), and VTT-R-95EE-H_2_O (97 g).

### 2.3. Isolation of Resveratrol Tetramers and Fingerprinting Analysis of Extracts and Fractions

The isolation and purification of resveratrol tetramers with the same molecular mass of (+)-hopeaphenol, (+)-vitisin A, and (−)-vitisin B from VTT-R-95EE-EA were according to the previous reports [32]. Briefly, the VTT-R-95EE-EA was separated by a silica gel column (150 cm × 12 cm) and then eluted with increasing polarity of solvent mixtures of *n*-hexane and EA to obtain twenty-five fractions. Fraction 24 was separated by a RP-18 column (5 cm × 60 cm) and then eluted with solvent mixtures of water and methanol from 1/1 to 0/1 to obtain eight fractions of fraction 24A to fraction 24H (each 100 mL). The (+)-hopeaphenol and (+)-vitisin A were obtained by semi-preparative HPLC from fraction 24E, and (−)-vitisin B was obtained by semi-preparative HPLC from fraction 24F. These three resveratrol tetramers could also be isolated and purified from VTT-R-95EE-EA [32] or VTT-S-95EE-EA [33]. The HPLC fingerprinting analysis included two extracts, VT-95EE and VTT-R-95EE, and two partitioned fractions, VT-95EE-EA and VTT-R-95EE-EA. The analytical Phenomenex Luna C-18 column (5 μm, 250 mm × 4.6 mm) was performed using Hitachi L-7000 chromatography system with a gradient elution program was set in solvent mixtures of distilled water and acetonitrile as follows: water/acetonitrile, 95/5, at 0 to 5 min; 76/24, at 35 min; 60/40, at 70 min. The 10 mg/mL of each sample was prepared, and 20 μL was injected for analysis. The flow rate of mobile phase was 1.0 mL/min. The column temperature was kept at room temperature, and the wavelength was set at 280 nm for monitoring. The identified compounds included resveratrol (35.63 min), (+)-hopeaphenol (resveratrol tetramer, 42.07 min), (+)-ampelopsin C (resveratrol trimer, 42.71 min), (+)-ε-viniferin (resveratrol dimer, 46.85 min), (+)-vitisin A (resveratrol tetramer, 50.83 min), and (−)-vitisin B (resveratrol tetramer, 61.01 min) based on the elution sequence [31,32,33,34].

### 2.4. AChE Inhibitory Activities In Vitro

The synthetic acetylthiocholine iodide was used as substrates for in vitro AChE activity assays [19]. Each extract, partitioned fraction (100 μg/mL), or purified resveratrol and its tetramers (resveratrol, 5, 10, 40, and 80 μM; hopeaphenol, 1, 5, 10, and 40 μM; vitisin A and vitisin B, 0.25, 0.5, 1, 2, and 4 μM) (dissolved in DMSO) was pre-mixed with the diluted recombinant human AChE (0.079 μg/mL in 100 mM phosphate buffer, pH 7.5) for 5 min, and then acetylthiocholine iodide was added. The released thiocholine was reacted with DTNB to generate yellowish 5-thio-2-nitrobenzoate. The donepezil was used as the positive control for comparisons. The absorbance changes were recorded at 405 nm for 10 min. The DMSO instead of sample solution was used as the blank, and the absorbance value at 405 nm was expressed as 100% AChE activity. The AChE inhibitory activities (%) were as following: [(A405_blank_ − A405_sample_)/(A405_blank_)] × 100%. The concentration for ACE 50% inhibition (IC_50_) was calculated from each linear equation from three used concentrations for AChE inhibitions. For resveratrol and hopeaphenol, 5, 10, and 40 μM were used; for vitisin A and vitisin B, 0.5, 1, and 2 μM were used.

### 2.5. MAO-B Inhibitory Activities In Vitro

The kynuramine was used as the MAO-B substrate to generate a fluorescent product, 4-hydroxyquinoline, which was used to screen MAO-B inhibitory activity assays [35]. The total 100 μL solution contained 2 μL MAO-B (2.5 U/mL), the different concentrations of resveratrol and its tetramers in DMSO (resveratrol, 20, 40, 60, and 80 μM; hopeaphenol, 50, 100, 150, and 200 μM; vitisin A, 2.5, 5, 10, and 20 μM; vitisin B, 0.625, 1.25, 2.5, and 5 μM), and 40 μL of 200 mM kynuramine (dissolved in 10 mM phosphate buffer, pH 7.5) for 30 min, and the reaction was stopped by adding 0.4 M perchloric acid and 1 M NaOH. The deprenyl was used as the positive control. The fluorescence ratio (F) of Ex_315 nm_/Em_380 nm_ was recorded. The DMSO instead of sample solution was used as the blank, and the calculated fluorescence ratio of Ex_315 nm_/Em_380 nm_ was expressed as 100% MAO-B activity. The MAO-B inhibition (%) was calculated as the following: [(F_blank_ − F_sample_)/(F_blank_)] × 100%. The IC_50_ of MAO-B activity was calculated from each linear equation from three used concentrations for MAO-B inhibitions. For resveratrol, 20, 40, and 60 μM were used; hopeaphenol, 50, 100, and 150 μM were used; for vitisin A, 2.5, 5, and 10 μM were used; vitisin B, 0.625, 1.25, and 2.5 μM were used were used.

### 2.6. Effects of Resveratrol and Resveratrol Tetramers on Neuroprotective Activities against MGO-Treated SH-SY5Y Cells

The neuroprotective effects of resveratrol and resveratrol tetramers, including hopeaphenol, vitisin A, and vitisin B, on MGO-treated human SH-SY5Y neuroblastoma cells were following the report [36] with some modifications. The SH-SY5Y cells were cultured at 37 °C in the mixed medium of EMEM/F12 (1:1) and 10% FBS in the humidified atmosphere and 5% CO_2_. The SH-SY5Y cells (3 × 10^4^ cells/well) were seeded in the 96-well plate overnight, and then resveratrol and resveratrol tetramers (2.5, 5, 10, and 20 μM) or 0.1% DMSO (the blank and the control) were added for a 24-h incubation. The treated SH-SY5Y cells were washed with PBS, and then the media without (the blank) or with 500 μM MGO (the control group and sample groups) were added for additional 24-h incubation at 37 °C. The cell viability was assayed by the SRB dyes [28] as follows. The MGO-treated SH-SY5Y cells were washed twice with PBS, fixed by 10% trichloroacetic acid, and then stained with 0.2% SRB (dissolved in 1% acetic acid) for 30 min. The unbound dyes were removed by washing the stained cells with 1% acetic acid. The bound dyes in alive cells were extracted by 500 µL of 10 mM Tris-HCl buffer (pH 7.9). The absorbance at 540 nm was measured using an ELISA reader (TECAN, Männedorf, Switzerland). The cell viability recovered by resveratrol and its tetramers was expressed in percentages relative to the blank (%).

### 2.7. Treatments of VT-95EE or VT-95EE-EA or Purified Compounds of (+)-Vitisin A or (−)-Vitisin B in Scopolamine-Induced Amnesiac ICR Mice

Treatments of VT-95EE or VT-95EE-EA or purified compounds of (+)-vitisin A or (−)-vitisin B, in scopolamine-treated ICR mice were following the previous report [19]. The protocols of the present animal experiments were reviewed and approved (LAC-2016-0218), and housed in the Laboratory Animal Center, Taipei Medical University. The 6-week-old male ICR mice were from the Laboratory Animal Center, College of Medicine, National Taiwan University (Taipei, Taiwan). The Prolab^®^ RMH2500 normal diets (LabDiet Co., St. Louis, MO, USA) and water were free access during the animal experiments. After one week of acclimation, the VT-95EE or VT-95EE-EA pre-treatments were used to evaluate preventive functions as follows. There were seven groups and each group contained 6 heads of ICR mice, including the blank, the control, the VT-95EE (200, 400 mg/kg) groups, the VT-95EE-EA (200, 400 mg/kg) groups, and donepezil group (the positive control). At the first stage, the extract (VT-95EE and VT-95EE-EA)-administered group, ICR mice were treated from day 1 to day 11 by a single oral administration daily by gavage. Meanwhile, mice were treated from day 1 to day 11 by an equal volume of distilled water in the blank, the control, and the donepezil group once a day by gavage. At the second stage, from day 12 to day 15, the same treatment procedure as the day 1 to day 11 was performed in the groups of VT-95EE, VT-95EE-EA, the blank, and the control daily by gavage; mice in the donepezil group were treated with donepezil (5 mg/kg) daily. Except for the blank, each mouse in all groups was intraperitoneally injected with scopolamine (1 mg/kg) 30 min after the oral administration. Each mouse in the blank group was injected with an equal volume of PBS instead of scopolamine. Each mouse received a learning and memory evaluation by a passive avoidance test 30 min after scopolamine injection at day 13 to day 14. All mice were sacrificed in the end of animal experiments at day 15, and the brain tissue of mouse in each group was isolated, weighted, and stored at −80°C for further uses.

Two resveratrol tetramers of (+)-vitisin A or (−)-vitisin B were used to evaluate the active components of VT-95EE-EA in scopolamine-treated amnesiac ICR mice. The pre-treatment was used to evaluate preventive functions. There were five groups and each group contained 6 heads of ICR mice, including the blank, the control, the vitisin A (40 mg/kg) group, the vitisin B (40 mg/kg) group, and the donepezil group (the positive control). At the first stage, the purified compound (vitisin A and vitisin B) administered group, ICR mice were treated from day 1 to day 9 by a single oral administration daily by gavage. Mice were treated from day 1 to day 9 an equal volume of distilled water in the blank, the control, and the donepezil group once a day by gavage. At the second stage, from day 10 to day 13, the same treatment procedure as the day 1 to day 9 was performed in the groups of (+)-vitisin A, or (−)-vitisin B, the blank, and the control daily by gavage; mice in the donepezil group were treated with donepezil (5 mg/kg) daily. Except for the blank, each mouse in all groups was intraperitoneally injected with scopolamine (1 mg/kg) 30 min after the oral administration. Each mouse in the blank group was injected with an equal volume of PBS instead of scoploamine. Each mouse received a learning and memory evaluation by a passive avoidance test 30 min after scopolamine injection at day 11 to day 12. All mice were sacrificed in the end of animal experiments at day 13, and the brain tissue of mice in each group was isolated, weighted, and stored at −80 °C for further uses.

### 2.8. Learning and Memory Evalustions by Passive Avoidance Tests of Amnesiac ICR Mice

The improvements of impaired learning and memory function of amnesiac mouse by treatments of VT-95EE, VT-95EE-EA or purified compounds were evaluated by the passive avoidance test following the previous report [19]. The apparatus for passive avoidance test is a shuttle chamber composed of a bright box and a dark box; the dark box could be closed by a manual guillotine door (AccuScan Instruments Inc, Columbus, OH, USA). The dark box contained a wired metal floor, through which the electric foot shocks were automatically delivered in the acquisition trial. The first day was the acquisition trial, and the second day was the retention trial. For the acquisition trial, each mouse was placed around the shuttle chamber to acclimatize to the environment; 30 min later, each mouse was transferred to the bright box to record the time in the bright box (the step-through latency, s). If mouse entered the dark box, the guillotine door was manually closed, and an electric shock (0.3 mA) was delivered at the same time for 3 s to the feet of the entered mouse via the metal floor. For the retention trial, each mouse was run through the same protocol as the acquisition trial, and the step-through latency (s) in the retention trial was recorded for learning and memory function.

### 2.9. AChE Activities, MDA Levels, and Protein Expressions of Brain-Derived Neurotrophic Factor (BDNF) and Tropomyosin Receptor Kinase B (TrkB) in Mouse Brain Tissue Extracts

The isolated whole brain tissue was weighted, and powdered in liquid nitrogen using a mortar and pestle. The proteins in the tissue powders were suspended and extracted by 1 mL, one-fold-diluted PBS. The supernatants were saved by centrifuging extracts at 12,500× *g* for 60 min at 0 °C, and were packed into microcentrifuges and stored at −80 °C for further investigation. The protein contents in tissue extracts were determined by the BCA^TM^ protein assay kit (Thermo Fisher Scientific Inc.), and BSA was used to plot the standard curve. The AChE activity in the brain extracts was determined following the previous report [19]. The final concentration of donepezil at 100 nM was added to inhibit AChE activities in the tissue extracts, which the extracts of DTNB-reactable components were recognized as the sample blank in the AChE activity determination, and the AChE activity in the tissue extracts was expressed as [A405_sample_ − A405_sample blank_]/μg protein, and mouse in the blank was expressed as 100%. The malondialdehyde (MDA) levels in the brain extracts was determined using acidic TBA. The 320 μL of 0.5% TBA in 10% trichloroacetic acid was premixed with 80 μL of 5-fold diluted tissue extracts, and then heated at boiling water for 20 min, and then centrifuged at 10,000 rpm, 4 °C for 30 min. The supernatants were saved and the absorbance at 532 nm was determined and expressed as A532 nm/μg protein, and mouse in the blank was expressed as 100%. For determining protein expressions of BDNF and its receptor, TrkB, among brain extracts of the treated mice, the immune stains were performed onto polyvinylidene difluoride (PVDF) membranes (Millipore, Bedford, MA, USA). An equal amount of 40 μg protein of tissue extracts in different treated mice groups was separated by 10% sodium dodecyl sulfate-polyacrylamide gel electrophoresis. For immune stains, the primary antibody, including anti-BDNF antibody, anti-TrkB antibody, or anti-β-actin antibody, each was added in 1000-fold dilution by 1% gelatin in NaCl/EDTA/Tris (NET) solution, and incubated at 4 °C overnight; the horseradish peroxidase-conjugated IgG secondary antibody was added in 5000-fold dilution by 1% gelatin in NET solution, and incubated at room temperature for 2 h. The western chemiluminescent HRP substrate kit (WBKL S0050, Millipore) was used to stain the target protein expressions, which the image system [Syngene GeneGnome5 imaging system (Cambridge, UK)] was used to quantify the image levels of target protein in the immune blots.

### 2.10. Statistical Analyses

Data were expressed as mean ± SD from three independent quantitative experiments. The one-way analysis of variance (ANOVA) and the post hoc Tukey’s test were used to compare the differences among multiple groups in the step-through latency, AChE activity, or MDA levels in the brain extracts. It was considered as the significant difference (*p* < 0.05) among groups in which the marked different uppercase letters in the acquisition trial or the marked different lowercase letters in the retention trial, or the marked different lowercase letters in the AChE activity or MDA levels of the brain tissue extracts. The Student’s *t*-test was used to compare the neuroprotective activity of resveratrol or resveratrol tetramers in MGO-treated cells with the control. The different cell viability was considered as the significant differences when *p* < 0.05 (*), or *p* < 0.01 (**), or *p* < 0.001 (***). The GraphPad Prism 5.0 software (San Diego, CA, USA) was used to perform the statistical analyses.

## 3. Results

### 3.1. Effects of VT Extracts, Different Parts of VTT Extracts, VT- or VTT-Partitioned Fractions on AChE Inhibitory Activities

Figure 1A shows the effects of VT-HWE and VT-95EE at the same concentration of 100 μg/mL on the AChE inhibitory activities in vitro. At the same concentration, the VT-95EE showed 63.79% AChE inhibitory activity and VT-HWE showed 31.85% AChE inhibitory activity. The former showed almost 2-fold AChE inhibitory activities than that of VT-HWE. Therefore, three partitioned fractions of VT-95EE, naming VT-95EE-EA, VT-95EE-BuOH, and VT-95EE-H_2_O, were prepared and used to determine the differences of AChE inhibitory activities among partitioned fractions. It was found that the AChE inhibitory activities of VT-95EE-EA, VT-95EE-BuOH, and VT-95EE-H_2_O at the same concentration of 100 μg/mL were 81.94%, 27.82%, and 4.40%, respectively. The order of AChE inhibition was VT-95EE-EA > VT-95EE-BuOH > VT-95EE-H_2_O. Figure 1B shows the effects of HWE and 95EE of VTT different parts, naming VTT-S, VTT-L, and VTT-R, at the same concentration of 100 μg/mL on the AChE inhibitory activities in vitro. The VTT-S-HWE, VTT-L-HWE, and VTT-R-HWE showed 10.86%, 2.66%, and 4.01% AChE inhibitory activities, respectively. The VTT-S-95EE, VTT-L-95EE, and VTT-R-95EE showed 12.99%, 11.82%, and 67.93% AChE inhibitory activities, respectively. Among HWE and 95EE of VTT different parts, the VTT-95EE-R showed the highest AChE inhibitory activities and were used to perform liquid–liquid partitions. The AChE inhibitory activities of VTT-R-95EE-EA, VTT-R-95EE-BuOH, and VTT-R-95EE-H_2_O at the same concentration of 100 μg/mL were 74.04%, 24.94%, and 5.44%, respectively. The order of AChE inhibition was VTT-R-95EE-EA > VTT-R-95EE-BuOH > VTT-R-95EE-H_2_O. The VT-95EE showed similar anti-AChE activity to that of VTT-R-95EE, and VT-95EE-EA also showed similar anti-AChE activity to that of VTT-R-95EE-EA. Therefore, these potential AChE inhibitory extracts were assayed by HPLC fingerprinting analyses.

### 3.2. The HPLC Fingerprinting Analyses

Figure 2A shows the structures of resveratrol and resveratrol oligomers, including resveratrol dimer of (+)-ε-viniferin, resveratrol trimer of ampelopsin C, and resveratrol tetramers of (+)-vitisin A, (+)-hopeaphenol, and (-)-vitisin B, which were used to analyze the HPLC profiles of VT-95EE (B), VT-95EE-EA (C), VTT-R-95EE (D), and VTT-R-95EE-EA (E). The identified compounds included resveratrol (35.63 min, peak 1), (+)-hopeaphenol (resveratrol tetramer, 42.07 min, peak 2), (+)-ampelopsin C (resveratrol trimer, 42.71 min, peak 3), (+)-ε-viniferin (resveratrol dimer, 46.85 min, peak 4), (+)-vitisin A (resveratrol tetramer, 50.83 min, peak 5), and (-)-vitisin B (resveratrol tetramer, 61.01 min, peak 6) based on the elution sequence. All resveratrol and resveratrol oligomers were identified in the paired extracts of VT-95EE (Figure 2B) and VTT-R-95EE (Figure 2D) or the paired partitioned-fractions of VT-95EE-EA (Figure 2C) and VTT-R-95EE-EA (Figure 2E) with different patterns. Table 1 shows the area percentage (%) of identified resveratrol and resveratrol oligomers in extracts and partitioned fractions based on the HPLC chromatograms in the Figure 2. The VT-95EE (Figure 2B) and VT-95EE-EA (Figure 2C) exhibited high amounts of resveratrol (peak 1), (+)-ε-viniferin (resveratrol dimer, peak 4), and (+)-ampelopsin C (resveratrol trimer, peak 3), which accounted totally for about 94.5% and 88.84%, respectively, of total area. It was also found that the VTT-R-95EE (Figure 2D) and VTT-R-95EE-EA (Figure 2E) exhibited high amounts of (+)-hopeaphenol (resveratrol tetramer, peak 2), (+)-vitisin A (resveratrol tetramer, peak 5), and (−)-vitisin B (resveratrol tetramer, peak 6), which accounted for about 73.87% and 72.56%, respectively, of total area.

### 3.3. Effects of Resveratrol Tetramers on Inhibitory Activities of AChE and MAO-B and Neuroprotective Activities on MGO-Induced SH-SY5Y Cell Deaths

Figure 3A shows the AChE inhibitory activities of three resveratrol tetramers with the same molecular mass of vitisin A, hopeaphenol, and vitisin B in comparison with resveratrol. The donepezil is used as the positive control, which all test resveratrol-related compounds exhibited AChE inhibitory activities in a dose-dependent manner. The IC_50_ of donepezil, resveratrol, hopeaphenol, vitisin A, and vitisin B, respectively, was calculated to be 0.01, 17.72, 8.06, 1.29, and 1.33 μM in the present assays, in which the order of AChE inhibition was donepezil >> vitisin A ≅ vitisin B > hopeaphenol > resveratrol. Figure 3B shows the MAO-B inhibitory activities of three resveratrol tetramers with the same molecular mass of vitisin A, hopeaphenol, and vitisin B in comparisons with resveratrol. The deprenyl is used as the positive control, in which all test resveratrol-related compounds exhibited MAO-B inhibitory activities in a dose-dependent manner. The IC_50_ of deprenyl, resveratrol, hopeaphenol, vitisin A, and vitisin B, respectively, was calculated to be 2 × 10^−6^, 41.25, 88.16, 4.94, and 1.08 μM in the present assays, which the order of MAO-B inhibition was deprenyl >> vitisin B > vitisin A >> resveratrol > hopeaphenol. Figure 3C shows the neuroprotective activities of resveratrol and three resveratrol tetramers (2.5, 5, 10, and 20 μM) against 500 μM MGO-induced cell deaths in SH-SY5Y cell models. After being treated with 500 μM MGO, the SH-SY5Y cell viability was reduced from 97.33% (the blank) to 58.57% (the control). The pretreatments of resveratrol and hopeaphenol at concentrations of 2.5, 5, 10, and 20 μM showed no significant difference (*p* > 0.05) compared to that of the control to elevate MGO-treated cell viability. The pretreatments of vitisin A (2.5, 5, 10, and 20 μM) or vitisin B (5, 10, and 20 μM) showed significantly to elevate MGO-treated cell viability compared to that of the control (*p* < 0.001, ***). Based on the present results, vitisin A and vitisin B showed better activities than resveratrol or hopeaphenol in anti-AChE and anti-MAO-B.

### 3.4. Effects of VT-95EE and VT-95EE-EA Pretreatments on the Improvements of Impaired Learning and Memory Functions in Scopolamine-Treated Amnesiac ICR Mice

Both VT-95EE and VT-95EE-EA exhibited the highest AChE inhibitory activities among extracts and fractions. Therefore, proof-of-concept of anti-AChE of VT-95EE and VT-95EE-EA were used to evaluate the improvements of dysfunctions of learning and memory in scoploamine-treated amnesiac ICR mice by passive avoidance tests. The treatment protocol of VT-95EE or VT-95EE-EA is shown in the Figure 4A. At the first stage, the pre-treatment of a single oral administration of VT-95EE or VT-95EE-EA (200 and 400 mg/kg) was performed daily in the first 11 days by gavage to highlight the preventive functions of VT extracts. Except extract-administered group, each mouse in all groups received a single, equal volume of distilled water by gavage daily in the first 11 days. At the second stage, from day 12 to day 15, except in the donepezil group, each mouse was treated with the same procedure as the first 11 days; in the donepezil group, the mice were treated with 5 mg/kg donepezil daily. Except for the blank, each mouse in all groups was intraperitoneally injected with scopolamine (1 mg/kg) 30 min after the oral administration. Each mouse in the blank group was injected with an equal volume of PBS instead of scopolamine. Figure 4B shows results of the step-through latency (s) of treated mice in the passive avoidance test. In the acquisition trial, the step-through latency (s) of treated mice and mice in the blank showed no significant difference (*p* > 0.05) among the groups. In the retention trial, the step-through latency (s) of mice in the extract-treated, donepezil-treated, or in the blank showed a significant longer time compared to those in the control (*p* < 0.05). These animal results showed that the treatment of VT-95EE and VT-95EE-EA could improve dysfunctions of learning and memory in scopolamine-treated amnesiac ICR mice. The dose effects of VT-95EE or VT-95EE-EA (200 and 400 mg/kg) were not found and showed no significant difference (*p* > 0.05) to enhancing the step-through latency in the retention trial, in which the extracts at dose of 200 mg/kg exhibited improved functions in learning and memory in scopolamine-treated amnesiac mice.

### 3.5. Effects of of Resveratrol Tetramers with the Same Molecular Mass of Vitisin A and Vitisin B Pretreatments on Impaired Learning and Memory Functions in Scopolamine-Treated Amnesiac ICR Mice

The resveratrol tetramers with the same molecular mass of vitisin A and vitisin B showed much higher anti-AChE and anti-MAO-B activities than those of resveratrol, and also exhibited neuroprotective activities to elevate cell viabilities against 500 μM MGO-induced SH-SY5Y cell deaths (Figure 3) and were selected for animal experiments. The treatment protocol of vitisin A and vitisin B (40 mg/kg) is shown in the Figure 5A. The pre-treatment was used to evaluate preventive functions. There were five groups and each group contained six heads of ICR mice, including the blank, the control, the vitisin A (40 mg/kg) group, the vitisin B (40 mg/kg) group, and the donepezil group (the positive control). At the second stage, from day 10 to day 13, the same treatment procedure as the day 1 to day 9 was performed in the groups of (+)-vitisin A, or (−)-vitisin B, the blank, and the control daily by gavage; mice in the donepezil group were treated with donepezil (5 mg/kg) daily. Except for the blank, each mouse in all groups was intraperitoneally injected with scopolamine (1 mg/kg) 30 min after the oral administration. Each mouse in the blank group was injected with an equal volume of PBS instead of scoploamine. Each mouse was received a learning and memory evaluation by a passive avoidance test 30 min after scopolamine injection at day 11 to day 12. Figure 5B shows the results of learning and memory evaluations by passive avoidance tests of treated amnesiac ICR mice. In the acquisition trial, the step-through latency (s) of the treated mice and mice in the blank showed no significant difference (*p* > 0.05) among groups. In the retention trial, the step-through latency (s) of mice in the vitisin A (40 mg/kg)-treated, donepezil-treated, or in the blank showed a significant longer time compared to those in the control (*p* < 0.05). The vitisin B (40 mg/kg)-treated amnesiac mice showed no significant difference (*p* > 0.05) compared to those in the control.

The changes of AChE activity, the MDA levels, and BDNF and TrkB protein expressions in the brain tissue extracts of amnesiac mice after vitisin A or vitisin B treatments are shown in the Figure 6, and donepezil treatment is used for comparisons. Mice in the scopolamine-treated control group showed increased AChE activity and MDA levels in brain tissue extracts, and had a significant difference compared to those in the blank (*p* < 0.05). Mice in vitisin A-treated and donepezil-treated groups showed a lower increase in AChE and MDA levels induced by scopolamine, and each had a significant difference compared to those in the control (*p* < 0.05) (Figure 6A,B). Mice in vitisin B-treated group showed lower increased AChE activities, and had a significant difference compared to those in the control (*p* < 0.05); however, no effect on reduction of MDA level was found (Figure 6A,B). Mice in the scopolamine-treated control group showed to reduce protein expressions of BDNF and TrkB in brain tissue extracts; however, mice in vitisin A-treated and donepezil-treated groups were showed to recover both protein expressions of the reduced BDNF and TrkB in brain tissue extracts of scopolamine-treated mice (Figure 6C). Mice in the vitisin B-treated group were shown to recover the reduced BDNF protein expressions, but not TrkB, in brain tissue extracts of scopolamine-treated mice (Figure 6C). Therefore, the vitisin A could be recognized as the active constituent in the preventive functions of VT extracts.

## 4. Discussion

The present study shows the first time in which pre-treatments of VT-95EE, VT-95EE-EA (Figure 4), and vitisin A (Figure 5) showed preventive functions in scopolamine-induced impaired learning and memory functions of amnesiac ICR mice, which were evaluated by passive avoidance test. Though the vitisin B and vitisin A exhibit similar anti-AChE and anti-MAO-B activities in vitro (Figure 3), the treatment of vitisin B to amnesiac mice (Figure 5) shows no effect on the increases of step-through latency in the retention trial of the passive avoidance compared to those of the scopolamine-treated control. The use of scopolamine to temporarily disrupt the neurotransmission by occupying the acetylcholine binding site in the muscarinic receptors induces amnesia, which is generally used in AD animal models and human clinical trials for developments of ACE inhibitors [13,14,15]. The behavior evaluation in passive avoidance test composed of an acquisition trial and a retention trial is a long-term emotional memory by stressful stimuli, in which the improved function is evaluated dependent on the effective communications between amygdala and hippocampus [37]. It is proposed that the differences of bioavailability and permeability of the blood-brain-barrier of the same molecular mass of vitisin A and vitisin B at the same dose treatment may be key factors to influence the outcome learning performance in amnesiac mice and need further investigations.

Mice in vitisin A-treated and donepezil-treated groups were shown to significantly (*p* < 0.05) lower the increased AChE and MDA levels, and also to elevate both protein expressions of BDNF and its ligand of TrkB induced by scopolamine compared to those in the scopolamine-treated control (Figure 6). The vitisin B-treated mice showed no ability to recover the reduced TrkB protein expressions in the brain tissues induced by scopolamine. It is proposed that the improvement of cognitive declines might be correlated with not only the AChE inhibition and reduced oxidative stress [4,5,6,7], but also modulations of synaptic plasticity by increasing BDNF/TrkB expressions [38]. BDNF is abundantly expressed in the hippocampus and cerebral cortex, and BDNF/TrkB signaling regulate brain synapses of short-term synaptic functions and long-term potentiation (LTP), which play protective roles against neuron cell apoptosis and hippocampus-mediated learning and memory functions [39,40,41]. Afzelin isolated from *Ribes fasciculatum* is shown to improve hippocampal LTP associated with the cognitive, learning, and memory behavior changes in scopolamine-treated mice c by increasing protein expressions of BDNF, and gene expressions of TrkB, AKT, and CREB [42]. Rutin treatment (100 mg/kg) reverses Aβ-induced neurotoxicity by increasing learning and memory functions compared to the control group evaluated by passive avoidance test, in which the gene expressions of ERK1, CREB and BDNF in the hippocampi of rats are increasing [43]. The hippocampus-dependent learning in the Morris water maze and passive avoidance tests is associated with a transient increase in BDNF mRNA expression in the hippocampus, and the use of anti-BDNF antibodies delivered into the rat brain continuously for 7 days with an osmotic pump causes impairment of memory by longer escape latencies in the water maze than those of IgG treatments [39]. It is reported that the TrkB mutant transgenic mice lost their learning ability compared to the littermate control mice in the complex or stressful learning paradigms, such as the simple passive avoidance test, radial maze and water maze [44]. The phosphorylated TrkB via BDNF binding is activated to regulate a variety of cellular processes in the nervous system, including Ras-PI3K-Akt signaling pathway for growth and survival of the neuron cells; GRB2-Ras-MAPK-Erk signaling pathway for neuronal differentiation; and modulation of phospholipase Cγ-mediated synaptic plasticity [45]. It is reported that the activity-dependent synaptic plasticity is calcium ion-dependent regulation of TrkB expression [46]. The different TrkB protein expressions in brain tissue extracts correlated with the learning behaviors in passive avoidance test between vitisin A-treated and vitisin B-treated mice may include intracellular neuronal calcium ion signaling, such as calcium ion stores and calcium ion influx associated mechanisms, which will need further investigation using cell models.

The vitisin A accounts for 2.5% and 24% of the identified resveratrol and resveratrol oligomers (Figure 2 and Table 1) in the HPLC chromatograms of the VT-95EE and VTT-R-95EE, respectively. The vitisin A shows potent inhibitory activities against AChE and MAO-B (Figure 3B,C), and also exhibits neuroprotective activities against MGO-induced cell deaths (Figure 3C). The vitisin A may be recognized as a marker constituent, and one active constituent in VT-95EE or VTT-R-95EE, which was shown synergistically with different active compounds to improve learning and memory functions in scopolamine-induced mice. The vitisin A has been reported to exhibit the vasodilating effects on phenylephrine-induced tension by increasing NO releases from endothelium-intact thoracic aortic rings of SD rats, and also the antihypertensive effects on spontaneously hypertensive rats in which the greatest reduction in systolic blood pressure and diastolic blood pressure were 23.6 and 25.7 mmHg, respectively, at the 8-h after a single oral dose of 10 mg/kg [31]. The C57BL/6 mice treated with high fat diets together with vitisin A oral administration once a day at dose of 25 mg/kg were shown to reduce weight gains and the improved obesity-related cardiovascular risk parameters in plasma, including such as reduced total cholesterols, reduced total triglycerides, reduced low-density lipoproteins, and reduced free fatty acid [32]. Reagan-Shaw et al. [47] summarized the dose translations by normalizing body surface area between experimental animal treatments, and experimental animal treatments and clinical human trial. It is calculated by body surface area normalization that the human equivalent dose is about 3.24 mg/kg of human body weight based on vitisin A treatment of 40 mg/kg of mouse body weight, or that the human equivalent dose is 16.2 mg/kg of human body weight from VT-95EE treatment of 200 mg/kg of mouse body weight. An adult with average weight of 60 kg is suggested to consume 195 mg vitisin A or 973 mg VT-95EE per day to achieve the improving learning and memory functions in degenerative disorders. This shall be investigated further.

In conclusion, the vitisin A is shown to improve scopolamine-induced impaired learning and memory functions in amnesiac mice evaluated by the passive avoidance test. The vitisin A lowers the increased AChE activities and MDA levels and elevates protein expressions of the reduced BDNF and TrkB in the brain induced by scopolamine treatments. The vitisin A exists in both stem parts of the medicinal plant of *Vitis thunbergii* Sieb. & Zucc. (VT) and the root part of Taiwan endemic cultivar of *Vitis thunbergii* var. *taiwaniana* (VTT), and may provide resources for the treatment of degenerative disorders by development as a functional food for unmet medical needs.

## Figures and Tables

**Figure 1 biomedicines-10-00273-f001:**
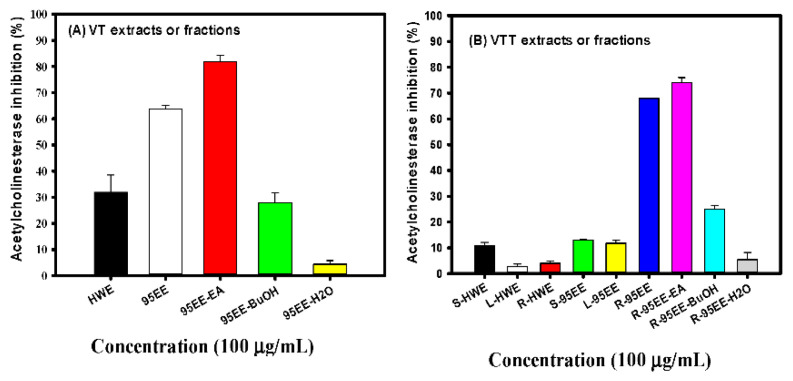
Effects of (**A**) VT-HWE, VT-95EE, VT-95EE-EA, VT-95EE-BuOH, and VT-95EE-H_2_O; and (**B**) HWE and 95EE of VTT-S, VTT-L, and VTT-R, and VTT-R-95EE-EA, VTT-R-95EE-BuOH, and VTT-R-95EE-H_2_O on acetylcholinesterase (AChE) inhibitory activities. The stem part of *Vitis thunbergii* Sieb. & Zucc. (VT) or the *Vitis thunbergii* var. *taiwaniana* (VTT) of different parts, root (VTT-R), stem (VTT-S), and leaf (VTT-L), each was extracted either by hot-water (HW) or 95% ethanol (95E) to get HW extracts (HWE) and 95EE. Data were expressed as mean ± SD of three independent quantitative experiments.

**Figure 2 biomedicines-10-00273-f002:**
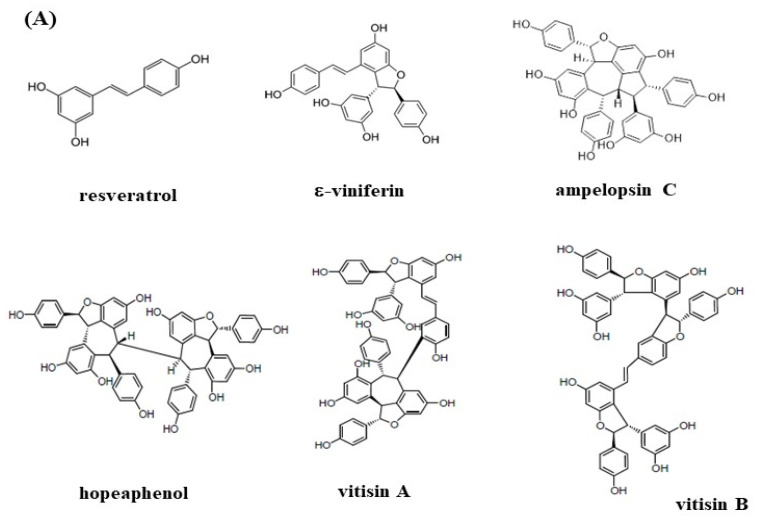
(**A**) The structures of the resveratrol and resveratrol oligomers, including resveratrol dimer of (+)-ε-viniferin, resveratrol trimer of ampelopsin C, and resveratrol tetramers of (+)-vitisin A, (+)-hopeaphenol, and (−)-vitisin B. The HPLC chromatograms of (**B**) VT-95EE; (**C**) VT-95EE-EA; (**D**) VTT-R-95EE; and (**E**) VTT-R-95EE-EA. The identified compounds included resveratrol (peak 1), (+)-hopeaphenol (resveratrol tetramer, peak 2), (+)-ampelopsin C (resveratrol trimer, peak 3), (+)-ε-viniferin (resveratrol dimer, peak 4), (+)-vitisin A (resveratrol tetramer, peak 5), and (−)-vitisin B (resveratrol tetramer, peak 6) based on the elution sequence.

**Figure 3 biomedicines-10-00273-f003:**
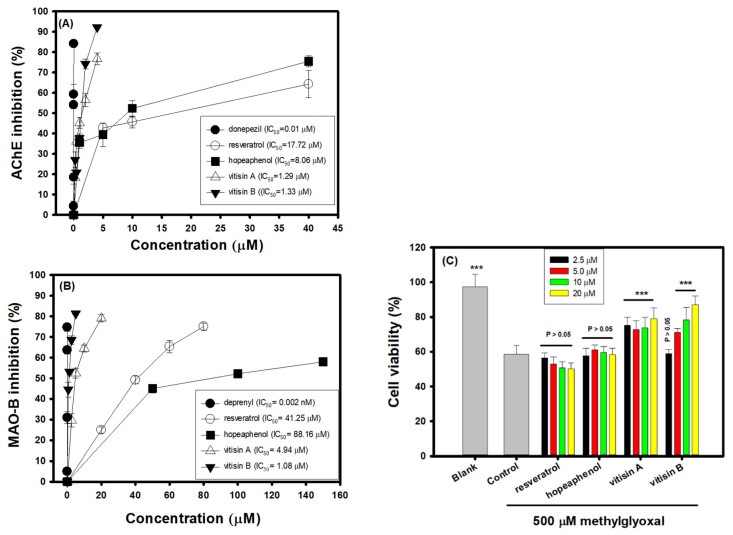
(**A**) Effects of the same molecular mass of resveratrol tetramers, vitisin A, hopeaphenol, and vitisin B, on acetylcholinesterase (AChE) inhibitory activities in comparisons with resveratrol, and the donepezil was used as the positive control; (**B**) Effects of the same molecular mass of resveratrol tetramers, vitisin A, hopeaphenol, and vitisin B, on monoamine oxidase (MAO)-B inhibitory activities in comparisons with resveratrol, and the deprenyl was used as the positive control; (**C**) Effects of resveratrol, vitisin A, hopeaphenol, and vitisin B, on neuroprotective activities (2.5, 5, 10, and 20 μM) in methylglyoxal-induced SH-SY5Y cell deaths. Data were expressed as mean ± SD of three independent quantitative experiments. The Student’s *t*-test was used to compare the neuroprotective activity of resveratrol or resveratrol tetramers in 500 μM methylglyoxal-treated SH-SY5Y cells (**C**) with the control. The different cell viability was considered as the significant differences when *p* < 0.001 (***).

**Figure 4 biomedicines-10-00273-f004:**
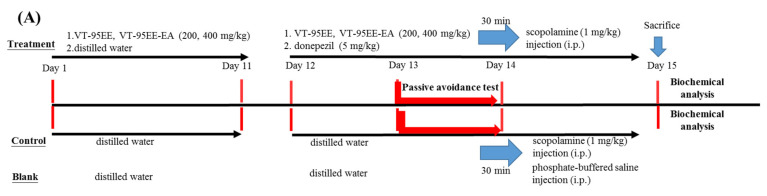
Effects of VT-95EE or VT-95EE-EA pretreatments (200 and 400 mg/kg) on the improvements of impaired learning and memory functions in scopolamine-treated amnesiac ICR mice. (**A**) The experimental protocol. There were seven groups (six heads of ICR mice/group), including the blank, the control, the VT-95EE (200, 400 mg/kg) groups, the VT-95EE-EA (200, 400 mg/kg) groups, and donepezil group (the positive control); (**B**) the learning and memory behaviors (the step-through latency, s) in the passive avoidance test. The scopolamine-injected mice without any sample treatments were used as the control. The one-way analysis of variance (ANOVA) and the post hoc Tukey’s test were used to compare the differences among multiple groups in the step-through latency (**B**). It was considered as the significant difference (*p* < 0.05) among groups which marked the different uppercase letters in the acquisition trial or marked the different lowercase letters in the retention trial.

**Figure 5 biomedicines-10-00273-f005:**
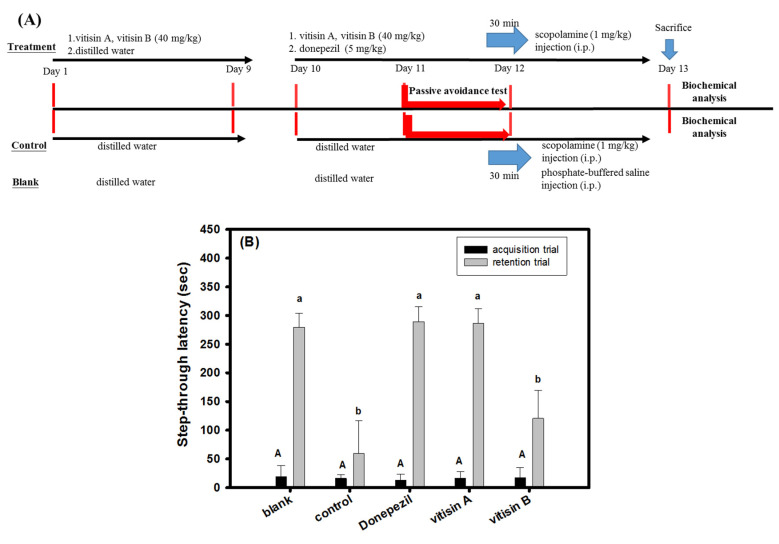
Effects of vitisin A or vitisin B pretreatments (40 mg/kg) on the improvements impaired learning and memory functions in scopolamine-treated amnesiac ICR mice. (**A**) The experimental protocol. There were five groups and each group contained 6 heads of ICR mice, including the blank, the control, the vitisin A (40 mg/kg) group, the vitisin B (40 mg/kg) group, and the donepezil group (the positive control); (**B**) the learning and memory behaviors (the step-through latency, sec) in the passive avoidance test. The scopolamine-injected mice without any sample treatments were used as the control. The one-way analysis of variance (ANOVA) and the post hoc Tukey’s test were used to compare the differences among multiple groups in the step-through latency (**B**). It was considered as the significant difference (*p* < 0.05) among groups which the marked different uppercase letters in the acquisition trial or the marked different lowercase letters in the retention trial.

**Figure 6 biomedicines-10-00273-f006:**
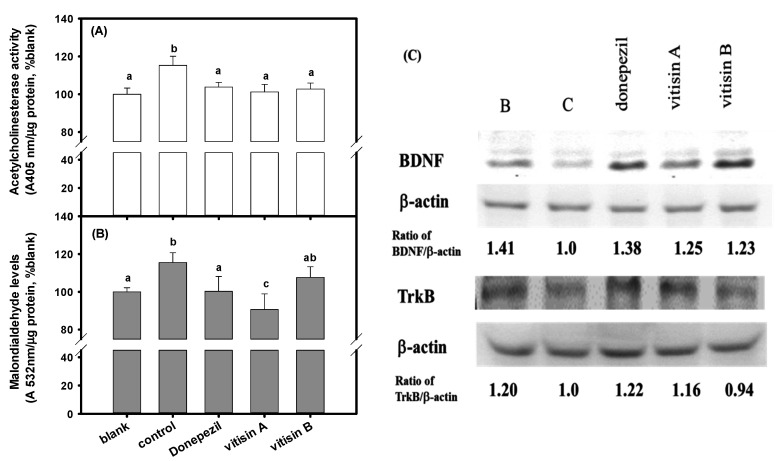
The changes of (**A**) AChE activity, (**B**) the MDA levels, and (**C**) BDNF and TrkB protein expressions, and expressed as ratios of BDNF/β-actin and TrkB/β-actin in the brain tissue extracts of amnesiac mice after vitisin A or vitisin B treatments (40 mg/kg). The scopolamine-injected mice without any sample treatments were used as the control. Data were expressed as mean ± SD of three independent quantitative experiments. The one-way analysis of variance (ANOVA) and the post hoc Tukey’s test were used to compare the differences among multiple groups. It was considered as the significant difference (*p* < 0.05) among groups which marked the different lowercase letters.

**Table 1 biomedicines-10-00273-t001:** The composition profiles of identified resveratrol and its oligomers in HPLC chromatograms of extracts and partitioned fractions ^#^.

		Extracts	VT-95EE	VT-95EE-EA	VTT-R-95EE	VTT-R-95EE-EA
	Area (%)					
Compound						
Monomer resveratrol (peak 1)			33.97 ^#^	33.90	0.41	0.76
Dimer (+)-ε-viniferin (peak 4)			47.89	44.43	0.86	0.96
Trimer (+)-ampelopsin C (peak 3)			12.64	10.51	1.10	1.12
Tetramer (+)- hopeaphenol (peak 2)			2.37	2.24	8.30	8.09
Tetramer (+)-vitisin A (peak 5)			2.52	2.30	24.37	19.33
Tetramer (-)-vitisin B (peak 6)			0.60	0.62	41.20	45.14

^#^ The composition profiles were based on the area in HPLC chromatograms of Figure 2.

## Data Availability

All figures and data used to support this study are included within this article.

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
