# Peer review of "Vitisin A, a Resveratrol Tetramer, Improves Scopolamine-Induced Impaired Learning and Memory Functions in Amnesiac ICR Mice"

_biomedicines, 2022, doi:10.3390/biomedicines10020273_

Round 1

Reviewer 1 Report

The pharmacology of novel plant medications belongs to the most interesting fields of the contemporary neuroscience. The research paper by Chen et al. reports  an improvement of learining and memory functions in mice treated with vitisin A, a resveratrol derivative isolated from smal-leaf grape Vitis thunbergii thaiwaniana roots.

Authors suggest that this natural substance may improve scopolamine induced memory impairment in mice by decrease of AChE and malondialdehyde levels. This makes the study potentially but cautiously valuable for the medical practice. The experimental paradigm is well considered experimental procedures such as SH-SY5Y neuroblastoma cell culture, estimation of MAO-B activity and behavioural tests  were kept good standard.  Appropriate statistical methods were applied with sufficient number of experimental data.

However, I have the following suggestions for the Authors:

Discussion

This section seems to look like a shorter, mainly repetitive and partly comparative description of the results. One can not find a complete mechanistic explanation of the effects observed. It is probably due to unclear receptor characteristics of the vitisin A molecule, however some further words dealing with potential biochemical or/and neurophysiological mechanisms of vitisin A action  e.g.  its affinity to some cellular targets should be provided. Are some histochemical/fluorescence studies dealing with resveratrol-derivatives available? They would probably be interesting for the readers in the context of changes revealed.

Figure 3. Description fonts within all internal frames of the graphs are poorly visible as well as too thin bars on the right. Please try to improve this.

Reviewer 2 Report

Please consider restructuring the abstract; it’s very confusing. Please keep in mind ‘you know what you know; I don’t know what you know’ make it simpler not the way it's reported in current form. Abstract should be background (also include rationale), method, results followed by summary all together. Riding on 95EE , VTT-R-95EE, VT-95EE etc. as a terminology from the beginning look unstructured.

You have a great amount of work, look complete experimentally.

Use a separate table for all the abbreviations.

Line 149: what is the catalog for 10 mesh sieves?

Please highlight how BDNF is involved in learning memory (mechanism)?

Why have passive avoidance tests been chosen?

Do you know anything about overexpression of BDNF (any patch clamp data) changes electrode potentials of cells? Is there any role of ion channels such as KV channel proteins in the learning process?

Does VT-95EE or VTT-R-95EE have any vasodilation effect?
